# Changes in Alienation in Physical Education Classes, School Happiness, and Expectations of a Future Healthy Life after the COVID-19 Pandemic in Korean Adolescents

**DOI:** 10.3390/ijerph182010981

**Published:** 2021-10-19

**Authors:** Seung-Man Lee, Jung-In Yoo, Hyun-Su Youn

**Affiliations:** 1Department of Physical Education, College of Education, Korea University, Seoul 02841, Korea; lsm14pe@korea.ac.kr; 2Department of Physical Education, College of Education, WonKwang University, Iksan-si 54538, Korea

**Keywords:** adolescents, alienation in PE classes, school happiness, future healthy life expectations, COVID-19 pandemic

## Abstract

This study aims to investigate the changes in the structural relationship between alienation in physical education (PE) classes, school happiness, and future healthy life expectations in Korean adolescents after the COVID-19 pandemic. The data were collected from Korean adolescents using different scales. The collected data were analyzed using frequency analysis, reliability analysis, validity analysis, independent t test, and path analysis. The key results were: First, there were partial changes in each of the parameters since the outbreak of COVID-19. Second, before the pandemic, alienation in PE classes negatively affected school happiness, and school happiness positively affected expectations of a future healthy life; however, alienation in PE classes did not affect the expectations of a future healthy life, showing a complete mediating effect. Third, during the pandemic, alienation in PE classes negatively affected school happiness, and school happiness positively affected the expectations of a future healthy life; alienation in PE classes negatively affected the expectations of a future healthy life, showing a partial mediating effect. These findings emphasize the importance and potential of school education, especially PE, in promoting happiness and healthy lives in adolescents. We expect these findings to have practical implications for future research by presenting theoretical and empirical data.

## 1. Introduction

In response to the first confirmed case of the coronavirus disease 2019 (COVID-19) in South Korea in January 2020, the government implemented various measures to combat the outbreak. In addition to direct medical responses, the government proposed indirect response measures that involved the public, such as wearing a face mask and practicing social distancing (Korean Ministry of Health and Welfare, 2020 [1]). However, there have been claims that despite the benefits of the nationwide use of face masks and social distancing practices to prevent the spread of the infection, they also prompt depression, anxiety, loneliness, and reduce physical activities. With the pandemic being prolonged for nearly a year and six months, public health concerns have become a tangible threat. Recent studies report that the increasingly prevalent socially isolated lifestyle resulting from the pandemic is adversely impacting people’s health. The new regular, established by lockdowns during the pandemic, including work from home, increased use of digital devices, and diminished willingness to exercise, has led to mental, physical, and social health problems [2,3,4]. Some prime examples of adverse health impacts are weight gain, poor posture, irregular sleep, stress and anxiety, and vitamin D deficiency [5,6,7].

Studies show that life restrictions during the COVID-19 pandemic affected adults and severely impacted the health of middle school students with respect to their rapid physical growth [8,9,10]. Recent studies on adolescents’ health during the COVID-19 pandemic document such phenomena. More specifically, they reported that the COVID-19 pandemic has taken a toll on adolescents’ mental health, contributing to elevated depression, anxiety, social isolation, maladaptive behavior, and stress [11,12,13,14,15]. Furthermore, studies also note that adolescents have experienced a decline in their physical health. COVID-19 led to a spike in the prevalence of obesity among adolescents [16], diminished participation in physical activities [17,18], and lowered their levels of physical activity involvement [19]. In addition, adolescents have shown a deterioration in health-related quality of life (HRQoL) [20] and poorer lifestyle habits [17,18,21].

However, these studies are limited because they focused solely on the physical and mental aspects of adolescents’ health during the COVID-19 pandemic. Currently, there is a consensus on the notion that adolescents’ health is not concretely divided into physical and mental aspects [22,23]. While active physical activity leads to positive mental health [24], social relationships with peers blend with these factors, thus impacting adolescents’ overall health [25]. For this reason, adolescents’ social health is an important topic of interest during the COVID-19 pandemic. In particular, because adolescents spend most of their day on online and offline school classes, their social health should be examined concerning their school lives or classes [26]. Thus, we aimed to explore adolescents’ social health during the COVID-19 pandemic based on physical education (PE) classes in school, when students are most active, and their peer relationships and activities are most evident [27]. In particular, we established alienation in PE classes, school happiness, and future healthy life expectations as elements of adolescents’ social health to analyze the structural relationships among these parameters before and during the COVID-19 pandemic. 

The first parameter, “alienation,” refers to a state in which students feel distant or severed from the overall PE class atmosphere and thus develop negative perceptions and feelings [28]. Alienation in PE classes is caused by various factors, including teachers, course content, peer relationships, PE facilities, and PE uniformity [29]. In addition, repetition of boring or meaningless activities [30,31], inappropriate interaction with the teacher [28,32,33,34], competitive atmosphere [35], and poor interpersonal relationships due to low self-concept or self-esteem have also been identified as some of the major causes [28,30,33,34,36].

The second parameter, “school happiness,” is a cognitive and affective factor describing students’ beliefs about themselves and their evaluation of their comprehensive school life based on what they experience during learning and in their interpersonal relationships [37]. School happiness encompasses concepts such as perceived well-being, quality of life, and life satisfaction [38].

The third parameter, “future healthy life expectations,” is the anticipated satisfaction of a healthy life in the future. In other words, it refers to one’s hope or expectation that their future life would be more satisfactory [39]. This is a concept related to questions such as “Will I have a healthy life in the future?” “Will my future be happy or not?” “Will my beliefs and thoughts come true?” “What type of person will I become in the future?” “How will the world be changed in the future?” For adolescents who will be future leaders, such life expectations are more important than others. Future healthy life expectations have been documented by many studies to be deeply associated with various factors of individuals’ internal traits [40].

The above-described variables have interconnectivity; it is reported that students perceive inappropriate companionship, lack of moral and emotional empathy, and psychological burden due to alienation occurring in the physical education class situation [41,42]. This can negatively affect students’ satisfaction with school life, and it can be expected that their happiness decreases due to these negative effects. In addition, it is reported that the sense of happiness perceived in school affects students’ school life, health problems, and future life [43]. In this context, in order to prepare for the COVID-19 pandemic situation and the post-COVID-19 era, empirical research with a focus on physical education classes and school happiness is needed to improve the future life expectations of socially healthy teenagers. This can be said to be of great significance in terms of understanding the detailed relationship between various predictive variables that affect future life expectations. 

Adolescence is a period wherein individuals prepare for their independent lives as adults, and school is where they enhance their social health and cultivate their dreams. Therefore, this study investigates the structural relationships among alienation in PE classes, school happiness, and future healthy life expectations among adolescents to examine the changes in their social health since the outbreak of COVID-19. 

Hence, we developed the following study questions: First, are there changes in the study parameters since the outbreak of COVID-19? Second, are there changes in the structural relationships among the study parameters since the outbreak of COVID-19?

## 2. Materials and Methods

### 2.1. Study Population

In this study, using the same sample, we repeatedly compared and measured from the changes in the structural relationships among alienation in PE classes, school happiness, and future healthy life expectations in adolescents after the outbreak of COVID-19 compared to the pre-COVID-19 period. The study population consisted of students who participated in PE in school and were purposively sampled according to statistical studies by Yang [44], Kim and Cha [45], and other relevant books.

A pilot survey was conducted with PE students to evaluate the content validity of the questionnaire before the primary survey. After securing the validity of the questions through preliminary surveys and meetings with experts, the first data survey was performed on 400 middle and high school students who were conducting offline P.E. classes (Physical education classes where students go to school and learn sports-related contents in playgrounds, gymnasiums, classrooms) before the pandemic (October 2019). Of the 400 sampled participants, 390 completed the questionnaire, and after excluding 15 questionnaires with careless responses or missing responses, 375 were included in the analysis. The second data (August 2021) survey during the COVID-19 pandemic was conducted on 400 middle and high school students who were conducting parallel blended PE classes (PE classes where students learn through the web at home on days that they do not go to school, and physical education-related contents on days they go to school, in playgrounds, gymnasiums, classrooms). Of the 400 sampled participants, 360 completed the questionnaire, and after excluding 56 questionnaires with careless or missing responses, 304 were included in the analysis. This study was conducted in accordance with the guidelines of the Declaration of Helsinki and based on the approved IRB (P01-202006-22-001), the study subjects were sampled except for vulnerable and disabilities students. The demographic characteristics of the participants are presented in Table 1. 

### 2.2. Instruments

We used the following instruments to test the structural relationships among alienation in PE classes, school happiness, and future healthy life expectancies in adolescents. The questionnaire was divided into four sections: demographics, alienation in PE classes, school happiness, and future healthy life expectations. All instruments were validated in previous studies. The instruments used are described below.

#### 2.2.1. Alienation in PE

Adolescents’ alienation in PE was measured using the instrument developed by Kim [29] and validated by Hwang and Lee [46], Seo, Lee, and Song [47]. This is a 28-item scale comprising six domains (alienation in curriculum, alienation in athletic competence, alienation in peer relationships, alienation in PE facilities, and alienation in PE uniform) that uses a five-point Likert scale. The internal consistency, as measured by Cronbach’s α, was 0.816–0.923, indicating high reliability.

#### 2.2.2. School Happiness

Adolescents’ school happiness was measured using the instrument developed by Kim and Kim [48] and validated by Lee [49]. This 24-item scale comprises six domains (peer relationships, relationships with teachers, self-efficacy, environmental satisfaction, pleasure in learning activities, and psychological stability) that use a five-point Likert scale. The internal consistency, as measured by Cronbach’s α, was 0.866–0.850, indicating high reliability.

#### 2.2.3. Future Healthy Life Expectations 

Adolescents’ future healthy life expectancies were measured using the instrument developed by Noh [50] and Kim and Kim [49] after modifying the scale for use in this study. This is a 16-item scale comprising four domains (interpersonal relationship, economic life, healthy life, and self-realization) that uses a five-point Likert scale. The internal consistency, as measured by Cronbach’s α, was 0.836–0.929, indicating high reliability.

Confirmatory factor analysis (CFA) performed to test the fit of the scales confirmed that all the fit indices were above the mark of less than 0.8 for RMR, 0.9 or higher for GFI, NFI, IFI, TLI, and CFI, and 1.0 for RMSEA: x^2^ = 381.956, DF = 51, GFI = 0.909, CFI = 0.931, NFI = 0.921, RMR = 0.052, and RMSEA = 0.098.

### 2.3. Data Collection

The data were collected through an in-person questionnaire survey before the COVID-19 outbreak and an online survey after. Before the survey, we contacted the PE teacher at the study school and obtained informed consent. Then, we visited the school to explain the study’s purpose and other details, and data were collected from those who consented to participate in the survey via a self-report method. The survey took less than 10 minutes to complete, and careless responses or missing values were excluded from the analysis. The second survey round conducted during the COVID-19 pandemic was administered online. Information and instructions for the study were explained in detail to obtain the most honest responses. The survey took less than 10 minutes to reflect on, with regard to the particular context of an online survey. Consent was obtained from the school principal and students’ legal guardians before the first and second rounds of surveys. 

### 2.4. Data Analysis

Data collected through two rounds of the survey were analyzed using SPSS 18.0 (IBM Corp., Armonk, NY, USA) and AMOS 18 software (IBM Corp., Armonk, NY, USA). First, the reliability and validity of the scales used in this study were tested using internal consistency Cronbach’s α and CFA. Second, demographic characteristics were analyzed using frequency analysis. Third, the changes in the study parameters since the outbreak of COVID-19, compared to before, were analyzed using independent t tests. Finally, the structural relationships among the parameters before and during the COVID-19 pandemic were analyzed using path analyses. Statistical significance was set at *p* < 0.05.

## 3. Results

### 3.1. Changes in the Study Parameters during the COVID-19 Pandemic Compared to the pre-COVID-19 Period

Table 2 shows the results of the independent t tests performed to analyze the changes in alienation in PE classes, school happiness, and future healthy life expectations in Korean adolescents since the outbreak of COVID-19. Compared to before, there were partial changes in these factors during the COVID-19 pandemic. First, alienation in PE facilities was lower during the COVID-19 pandemic (2.95 ± 1.17) than pre-COVID-19 period (3.17 ± 1.29). Alienation in PE uniforms was lower during the COVID-19 pandemic (3.23 ± 1.14) than in the pre-COVID-19 period (3.23 ± 1.14). However, there were no changes in the alienation in athletic competence, alienation in curriculum, and alienation in peer relationships during the COVID-19 pandemic from before. Second, the relationship with the teachers’ domain of school happiness was higher during the COVID-19 pandemic (3.69 ± 0.76) than pre-COVID-19 period (3.52 ± 0.93). Furthermore, satisfaction with the class environment was higher during the COVID-19 pandemic (3.35 ± 0.89) than the pre-COVID-19 period (3.35 ± 0.89). However, there were no changes in the peer relationships, self-efficacy, and pleasure in learning activities during the COVID-19 pandemic from the pre-COVID-19 period. Finally, the economic aspect of future healthy life expectations was more pronounced during the COVID-19 pandemic (3.43 ± 0.83) than the pre-COVID-19 period (3.66 ± 0.94). However, there were no changes in interpersonal relationship, healthy life, and self-realization expectancies during the COVID-19 pandemic from before. 

### 3.2. Comparison of Structural Relationships among the Study Parameters between the pre- and during COVID-19 Periods

Table 3 shows the results of the path analyses performed to analyze the structural relationships among alienation in PE classes, school happiness, and future healthy life expectancies in Korean adolescents before the pandemic. The path model had a good fit: GFI = 0.920, CFI = 0.951, NFI = 0.935, RMR = 0.053, and RMSEA = 0.089. Kim (2005) [29] proposed that GFI, AGFI, NFI, TLI, and CFI of 0.8–0.9 or higher, RMR of 0.05–0.08 or lower, and RMSEA of 0.08 or lower indicate a good fit. Regarding our model, alienation in PE classes negatively affects school happiness (standardized regression coefficient = −0.449, t = −7.102). Second, alienation in PE classes did not negatively affect future healthy life expectations (standardized regression coefficient = −0.095, t = −1.695). Third, school happiness positively affects future healthy life expectations (standardized regression coefficient = 0.530, t = 8.425). In other words, there was a complete mediation wherein alienation in PE classes affected future healthy life expectations solely through school happiness.

Table 4 shows the results of the path analyses performed to analyze the structural relationships among alienation in PE classes, school happiness, and future healthy life expectancies in Korean adolescents during the pandemic. The path model had a good fit: GFI = 0.886, CFI = 0.923, NFI = 0.900, RMR = 0.049, and RMSEA = 0.098. Regarding our model, first, alienation in PE classes negatively affects school happiness (standardized regression coefficient = −0.492, t = −5.034). Second, alienation in PE classes negatively affects future healthy life expectations (standardized regression coefficient = −0.134, t = −2.212). Third, school happiness positively affects future healthy life expectations (standardized regression coefficient = −0.678, t = 9.430). In other words, there was a partial mediation wherein alienation in PE classes influenced future healthy life expectations directly and indirectly through the mediation of school happiness.

## 4. Discussion

This study aimed to investigate the structural relationships between alienation in PE classes, school happiness, and future healthy life expectations among Korean adolescents during the COVID-19 pandemic. It examines the changes in the parameters and their structural relationship compared to the pre-COVID-19 period to explore measures to enhance social health, an essential aspect of health in adolescence. We discuss our findings below. 

First, there were partial changes in the alienation in PE classes, school happiness, and future healthy life expectations among Korean adolescents since the outbreak of the COVID-19 pandemic. More specifically, the alienation of the PE facilities and uniform domains were lower during the pandemic than before. However, there were no changes in alienation in athletic competence, alienation in curriculum, and alienation in peer relationships between the two periods. These results are similar to previous findings of alienation in PE classes wherein it differs according to the educational and class environmental factors, such as PE facilities and PE uniforms [29]. In particular, one reason the degree of alienation regarding PE facilities and PE uniforms changed during the COVID-19 pandemic can be attributed to the possible bad weather, field condition, and conflict regarding the use of PE uniforms during in-person PE classes. However, with the implementation of hybrid PE classes (online and in-person) during the pandemic, students were less likely to wear a uniform, and the PE classes were less affected by the weather. Hence, measures to promote students’ school happiness should be devised considering that during the COVID-19 pandemic, adolescents spend most of their days completing online and offline schoolwork. This is one reason schools have continued with PE classes, even during the pandemic, by implementing hybrid designs. As previously described, various experiences acquired during PE classes affect students’ school happiness. Hence, schools should strive to adopt an ideal hybrid design, wherein online classes utilize humanistic and artistic approaches to PE, involving students in physical activities, while offline classes ensure they engage in the required physical activities that feature various contents and strategies while practicing social distancing and wearing a face mask. 

Second, the relationship with teachers and class environmental satisfaction domains of school happiness was higher during the pandemic than before. However, there were no changes in peer relationships, self-efficacy, and pleasure in learning activities between the two periods. These results are consistent with previous findings that interpersonal relationships with friends and teachers, academic achievement, and experiences of success in various subjects provoke happiness in students [48]. Conversely, it is similar to a previous report that concluded that students develop a negative perception of the PE class if they remain distant or feel severed from the class atmosphere and the PE teacher [51]. Further, a previous finding that students and teachers have different perceptions due to their power imbalance [52,53] supports our results. Specifically, PE teachers preferred and interacted more with highly athletic students or other specific groups of students in the in-person PE classes before the pandemic [53]. However, such phenomena were alleviated in the hybrid PE classes with online and in-person sessions. The physical distance between teachers and students in online classes seems to have helped students build a relationship of equality and interact with their PE teachers. Likewise, the physical distance between teachers and students in in-person classes in adherence to social distancing practices may have contributed to students’ perceived fairness in the instruction. Moreover, the conflict between teachers and students over wearing a PE uniform was partially curtailed in hybrid classes. 

Third, the economic aspect of future healthy life expectations was more pronounced during the COVID-19 pandemic than before, but there were no changes in the interpersonal expectations, healthy life expectations, and self-realization expectations domains. These results highlight that adolescence is a transitional period wherein individuals make critical decisions for their future. For this reason, adolescents have been reported to think positively about their future and set specific goals to carry on an independent life, even in adverse and vulnerable environments [54]. In addition, adolescents are primarily paying more attention to their academic performance to gain college admission than engaging in self-reflection and exploring career options. They are more focused on their future economic lives since the pandemic outbreak because they see many self-employed people experiencing financial hardship and other employed workers forced to take unpaid leave or losing their jobs due to the pandemic. We can infer that students who feel happy at school develop positive expectancies of their future lives based on their accomplishments, as suggested by a previous study reporting a positive association between school happiness and defined goal setting for the future [55]. Thus, during the COVID-19 pandemic, schools need to implement programs that promote students’ happiness. Schools have strived to engage students in learning through diverse hands-on experiences than simply imparting knowledge. Although such hands-on programs have been placed on hold due to the COVID-19 pandemic, creative online and in-person learning programs should be developed adhering to infection-prevention guidelines because relishing school enhances learning efficiency and increases students’ expectations for the future.

Finally, before the COVID-19 pandemic, school happiness had a complete mediating effect between alienation in PE classes and future healthy life expectations, wherein the former influenced the latter solely through school happiness. In contrast, during the pandemic, alienation in PE, both directly and indirectly, through the mediation of school happiness, influenced future healthy life expectancies. These results suggest that PE classes have had a more direct and significant impact on adolescents’ future lives since the outbreak of COVID-19 than before. In other words, while students who felt alienated in PE before the pandemic felt less happy at school, they could develop positive expectations for their future through the diverse curriculum and creative hands-on programs. However, since the pandemic outbreak, alienation in PE has influenced school happiness and their future lives. This is probably not exclusive to alienation in PE classes. With restrictions placed on their day-to-day lives during the pandemic, students link the microscopic phenomenon of alienation in PE classes to the macroscopic aspect of their future lives. Based on these results, schools should contemplate measures to reduce alienation in PE classes during the COVID-19 pandemic and continuously implement such strategies even after the pandemic ends, to promote future healthy life expectations in students. Ultimately, schools should also try to understand students’ circumstances through emotional bonding by identifying the causes of the alienation in PE classes during the COVID-19 pandemic. One good strategy is to assign individualized tasks or simple roles to such students during online and in-person classes to highlight their importance. The teacher needs to serve as psychological support for students, even during the COVID-19 pandemic, regardless of the class.

In summary, adolescents with low alienation in PE classes displayed a high level of school happiness and future healthy life expectations during the pandemic. This shows that social health has a positive impact on adolescents, even during the pandemic. Although many studies argued that adolescents’ physical and mental health is at risk [11,12,13,14,15,16,17,18,19,20,21], we observed that online and in-person hybrid PE classes promote adolescents’ social healthy by lowering alienation in PE and boosting school happiness and future healthy life expectations. To ensure adolescents’ social health during the COVID-19 pandemic, Kim’s [29] strategies to resolve the issue of alienation in PE should be utilized. This includes classes that: (1) allow students to enjoy various sports and improve their athletic competence; (2) allow students to learn about sports and experience the true pleasure of exercise; and (3) involve all students in sports and provide encouragement. Additionally, body consciousness and passion for sports proposed by Hwang and Lee [46] should be considered in PE programs that target the resolution of alienation.

Based on these findings and the limitations of this study, we present several suggestions for subsequent studies. First, this study was conducted with the adolescent population in the Republic of Korea during the COVID-19 pandemic. Hence, the findings cannot be generalized to other countries; subsequent studies should examine students from other countries. Second, adolescents develop future healthy life expectations as a result of various factors; therefore, subsequent studies should examine an array of potential predictors of future healthy life expectations based on our results. Third, although we used scales previously validated, future healthy life expectations is a relatively new concept; hence, studies should validate the scale using more diverse methodologies and assess its feasibility in various research. Fourth, we established the relationships among the study parameters based on existing literature. Thus, subsequent studies should establish the relationships among the parameters based on more diverse studies and theories. 

## 5. Conclusions

The key findings of this study are summarized as follows: First, there were partial changes in the alienation in PE classes, school happiness, and future healthy life expectations among Korean adolescents since the outbreak of COVID-19 compared to the time before. Specifically, the alienation in PE facilities and PE uniforms domains were lower during the COVID-19 pandemic than before, but there were no changes in alienation in athletic competence, alienation in curriculum, and alienation in peer relationships. Second, the relationship with teachers and satisfaction with class environment domains of school happiness were higher during the COVID-19 pandemic than before. However, there were no changes in peer relationships, self-efficacy, and pleasure in learning activities between the two periods. Third, the economic expectations of future healthy life expectations were higher during the COVID-19 pandemic than before, but there were no changes in the interpersonal relationships, healthy life, and self-realization expectancies. Finally, before the COVID-19 pandemic, school happiness completely mediated the influence of alienation in PE on future healthy life expectations. However, during the pandemic, school happiness only partially mediated the influence of PE directly and indirectly on future healthy life expectations. This suggests that adolescents were more dependent on school before COVID-19 than during the pandemic.

In conclusion, adolescence is a period when individuals grow into adults with desirable values by consistently interacting with various social factors and laying a foundation to design happy future lives. Hence, ensuring adolescents’ social, physical, and mental health through PE to help them gain comprehensive experiences and engage in positive and proactive thinking is crucial. Even during the COVID-19 pandemic, it is important to resolve the issues of alienation in PE classes such that adolescents can feel various forms of satisfaction at school and thus develop positive expectations for their future lives, as this would ultimately promote their social health. In light of these findings, the Office of Education and relevant organizations must develop and implement various educational programs to prevent and overcome alienation in PE. We believe that our study will serve as a valuable theoretical foundation for such efforts. 

## Figures and Tables

**Table 1 ijerph-18-10981-t001:** Participants’ demographic characteristics.

Variable	Category	First Round		Second Round	
		Number of Cases	Percentage (%)	Number of Cases	Percentage (%)
Sex	Male	180	48.0	165	54.3
	Female	195	52.0	139	45.7
School grade level	Middle school	109	29.1	172	56.6
	High school	266	70.9	132	43.4
Total		375	100	304	100

**Table 2 ijerph-18-10981-t002:** Changes of each study parameter since the outbreak of COVID-19 (Total possible score: 5.0).

Parameter		M ± SD		t	*p*
		Pre-COVID-19	During COVID-19		
Alienation in PE	Alienation in athletic competence	2.34 ± 1.05	2.35 ± 0.86	−0.100	0.921
	Alienation in curriculum	2.35 ± 1.05	2.46 ± 0.68	−1.606	0.109
	Alienation in peer relationship	1.99 ± 0.90	1.97 ± 0.72	0.215	0.830
	Alienation in PE facilities	3.17 ± 1.29	2.95 ± 1.17	2.325	0.020 *
	Alienation in PE uniform	3.23 ± 1.14	2.78 ± 1.03	5.359	< 0.001 ***
	Overall	2.61 ± 0.86	2.50 ± 0.67	1.883	0.060
School happiness	Peer relationships	4.00 ± 0.84	4.04 ± 0.86	−0.498	0.618
	Relationships with teachers	3.52 ± 0.93	3.69 ± 0.76	−2.538	0.011 *
	Self-efficacy	3.44 ± 0.79	3.38 ± 0.72	1.034	0.301
	Environmental satisfaction	3.35 ± 0.89	3.49 ± 0.76	−2.233	0.026 *
	Pleasure in learning activities	3.47 ± 0.88	3.59 ± 0.80	−1.771	0.077
	Overall	3.56 ± 0.73	3.64 ± 0.65	−1.493	0.136
Future healthy life expectations	Interpersonal relationship	4.15 ± 0.80	4.09 ± 0.73	1.004	0.316
	Economic life	3.66 ± 0.94	3.43 ± 0.83	3.274	0.001 **
	Healthy life	3.68 ± 0.95	3.55 ± 0.89	1.916	0.056
	Self-realization	4.04 ± 0.87	3.93 ± 0.78	1.698	0.090
	Overall	3.88 ± 0.79	3.75 ± 0.71	2.271	0.023 *

^***^ *p* < 0.001, ^**^ *p* < 0.01, ^*^ *p* < 0.05.

**Table 3 ijerph-18-10981-t003:** Structural relationships among the study parameters before the COVID-19 pandemic.

Hypothesis	Path			Standardized Regression Coefficient	S.E.	C.R.	*p*	Decision
1	Alienation in PE	→	School happiness	−0.449	0.051	−7.102	< 0.001 ***	Adopted
2	Alienation in PE	→	Future healthy life expectations	−0.095	0.058	−1.695	0.090	Rejected
3	School happiness	→	Future healthy life expectations	0.530	0.080	8.425	< 0.001 ***	Adopted

^***^ *p* < 0.001.

**Table 4 ijerph-18-10981-t004:** Structural relationships among the study parameters during the COVID-19 pandemic.

Hypothesis	Path			Standardized Regression Coefficient	S.E.	C.R.	*p*	Decision
1	Alienation in PE	→	School happiness	−0.492	0.140	−5.034	< 0.001 ***	Adopted
2	Alienation in PE	→	Future healthy life expectations	−0.134	0.085	−2.212	0.027 *	Adopted
3	School happiness	→	Future healthy life expectations	0.678	0.071	9.430	< 0.001 ***	Adopted

*** *p* < 0.001, * *p* < 0.05.

## Data Availability

The data presented in this study are available on request to the authors. Some variables are restricted to preserve the anonymity of the study participants.

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
