# Peer review of "Changes in Alienation in Physical Education Classes, School Happiness, and Expectations of a Future Healthy Life after the COVID-19 Pandemic in Korean Adolescents"

_ijerph, 2021, doi:10.3390/ijerph182010981_

Round 1
Reviewer 1 Report
The manuscript is well written and discussing and interesting and important impacts of COVID on adolescents, an understudied population during the pandemic. I had only minor comments. The dataset is rich, with multiple measurements pre and during COVID.
Introduction:
What about screen time as a mechanism, and time indoors? Not the focus of the study I realize but perhaps relevant to mention in passing here, or in the discussion/limitations?
It's unclear in the discussion of the three parameters whether students attended PE classes virtually or not at all.
Methods:
I like the use of "careless responses." This describes some adolescent's treatment of answering questionnaires perfectly!
These appear to be robust measures used, with many items per scale/dimension.
I missed a discussion of the demographic variables collected.
Results:
Again, I'm missing the description of the sample regarding demographics. The analysis says there were "frequency analyses" conducted about these also.
I'm unclear whether SEM or regressions are being run here. I presume the first. If so, this is a path (structural), without the prior confirmatory. What is the entire model combined together? (In other words, are these pathways tested controlling for the effects of the other parameters?). Sorry if this is already explained but it's unclear in the Methods/Results to this reviewer.
Discussion:
I missed a mentioning of study limitations here.
Reviewer 2 Report
In the present study authors investigated possible changes in the relationship between alienation in physical education classes, school happiness, and future healthy life expectancy in a large sample of Korean adolescents after the COVID-19 pandemic.
Overall, the study is quite interesting.
Please better explain the theoretical background of "alienation in physical education", "school happiness" and "future healthy life expectancy" constructs. It would be helpful for better understand the scope and results of the study.
The study is lacking in some qualitative information about the sample. Did you include or exclude participants with any neurodevelopmental (e.g. autism spectrum conditions, ADHD) or psychiatric condition (e.g. depression)? This could be a crucial variable to consider when interpreting the results especially in adolescents. Please also specify inclusion criteria.
Please critically discuss the implication of your findings.
Round 2
Reviewer 2 Report
The authors addressed all the point raised